# Surveillance to maintain the sensitivity of genotype-based antibiotic resistance diagnostics

**Allison L. Hicks** [1], **Stephen M. Kissler** [1], **Marc Lipsitch** [1,2‡], **Yonatan H. Grad** [1,3‡ *]

**1** Department of Immunology and Infectious Diseases, Harvard T. H. Chan School of Public Health, Boston, Massachusetts, United States of America, **2** Center for Communicable Disease Dynamics, Department of Epidemiology, Harvard T. H. Chan School of Public Health, Boston, Massachusetts, United States of America, **3** Division of Infectious Diseases, Department of Medicine, Brigham and Women's Hospital, Harvard Medical School, Boston, Massachusetts, United States of America

‡ These authors are co-senior authors on this work.
* ygrad@hsph.harvard.edu

**Data Availability Statement:** All data are publicly available in SRA/ENA (accession numbers provided in Table 1) or in referenced publications.

**Funding:** This work was supported by Grant U54GM088558 (Models of Infectious Disease

## Abstract

The sensitivity of genotype-based diagnostics that predict antimicrobial susceptibility is limited by the extent to which they detect genes and alleles that lead to resistance. As novel resistance variants are expected to emerge, such sensitivity is expected to decline unless the new variants are detected and incorporated into the diagnostic. Here, we present a mathematical framework to define how many diagnostic failures may be expected under varying surveillance regimes and thus quantify the surveillance needed to maintain the sensitivity of genotype-based diagnostics.

## Introduction

Antimicrobial resistance (AMR) poses a grave threat to global public health, underscoring the need for strategies to slow and control the spread of resistance. One direction is to develop fast and reliable diagnostics that minimize the delay between diagnosis and selection of an appropriate treatment regimen based on the target pathogen's antibiotic susceptibility profile [1,2]. A promising approach, use of pathogen genotype to predict AMR phenotype, has been facilitated by advances in rapid and cost-efficient amplification and sequencing. For example, the Cepheid GeneXpert MTB/RIF assay for rifampicin resistance in *Mycobacterium tuberculosis* and the SpeeDx ResistancePlus GC assay for ciprofloxacin resistance in *Neisseria gonorrhoeae* are already in clinical use, and many others are in the pipeline [3–5].

These genotype-based diagnostics must maintain high sensitivity to remain useful clinically. However, the emergence of novel resistance mechanisms will inevitably lead to a decline in sensitivity, perhaps exacerbated by variable prevalence of resistance determinants across populations [6]. Key to maintaining sensitivity is, therefore, sustained sampling and routine updating of the diagnostics with newly described resistance determinants. However, despite its importance for the structure of surveillance systems and, thus, for both public health agencies and diagnostics developers, the rate of sampling necessary for timely detection of novel resistance variants has been unclear.

Agent Study, Center for Communicable Disease Dynamics) from the National Institute of General Medical Sciences (ML) and Grant R01AI132606 from the National Institute of Allergy and Infectious Diseases (ALH, SMK, YHG). The funders had no role in study design, data collection and analysis, decision to publish, or preparation of the manuscript.

**Competing interests:** The authors have declared that no competing interests exist.

**Abbreviations:** AMR, antimicrobial resistance; bla$_{OXA}$, carbapenem-hydrolyzing class D beta-lactamase gene; GyrA, DNA gyrase subunit A; IS*Aba*1, *A. baumannii* insertion sequence 1; NAAT, nucleic acid amplification test; PBP2, penicillin binding protein 2; penA, penicillin binding protein 2 gene; qnr, Qnr family pentapeptide repeat protein gene.

Here, we use datasets of clinical isolates of multiple pathogens collected over 7–14 years to show that although the sensitivities of some genetic markers of resistance remain stably high, sensitivities of other markers rapidly decline because of the emergence of novel resistance variants. We present a simple mathematical framework that defines the rates of sampling and phenotypic testing necessary for early detection of novel resistance variants.

## Results

### Waning sensitivity of resistance markers

In the ideal scenario for a genotype-based antibiotic resistance diagnostic, phenotypic resistance is always encoded by a specific genotype—e.g., a single, stereotyped mutation or gene. To date, some combinations of bacteria and antibiotics approximately satisfy this criterion: target modification mutations in DNA gyrase subunit A gene (*gyrA*) maintain high sensitivity for predicting ciprofloxacin nonsusceptibility in *N. gonorrhoeae* and *Acinetobacter baumannii* (Fig 1A–1D). For other bacteria–antibiotic combinations, diagnostic genetic markers of resistance show decreased sensitivity over time, corresponding to increased incidence of previously rare or undetected resistance markers (Fig 1E–1J). The *gyrA* target modification mutation in *Klebsiella pneumoniae* isolates [7], for example, becomes a less sensitive predictor of ciprofloxacin nonsusceptibility as the incidence of isolates with acquired Qnr family pentapeptide repeat protein gene (*qnr*) genes (which code for target protecting proteins) increases (Fig 1E and 1F). Similarly, the emergence of the mosaic penicillin binding protein 2 gene (*penA*) (XXXIV) allele in *N. gonorrhoeae* clinical isolates [8] corresponds to decreased sensitivity of other target modification mutations for predicting penicillin nonsusceptibility (Fig 1G and 1H). Furthermore, decreased sensitivity of carbapenem-hydrolyzing class D beta-lactamase-58 gene (*bla*$_{OXA-58}$) for predicting imipenem nonsusceptibility in *A. baumannii* clinical isolates from the United States military healthcare system is associated with increased incidence of other oxacillinases (Fig 1I and 1J).

### Defining required sampling rate as a function of diagnostic failure threshold

Given the possible emergence of novel resistance variants, maintenance of a genotype-based AMR diagnostic requires surveillance and phenotyping of clinical specimens predicted to be susceptible, characterization of novel resistance determinants, and subsequent updating of the diagnostic. Once a resistant strain not captured by the current diagnostic test appears in the population, there is a simple relationship between the cumulative number of such cases and the probability that at least one will be detected: if *f* is the proportion of all genotypically susceptible cases that receive confirmatory phenotypic testing, and *N* is the number of variant cases, then the probability *x* that the new variant is detected in at least one of those cases is given by $x = 1-(1-f)^N$. Therefore, to have a probability of at least *x* that the new variant will be detected by the time *N* cases of it have occurred, the proportion undergoing confirmatory testing must be

$$f \geq 1 - (1-x)^{1/N}$$

Thus, to be 95% (*x* = 0.95) confident that a novel variant is detected by the time it has occurred in a total of 100 (*N*) cases, a sampling fraction (*f*) of approximately 0.03 is required (i.e., 3% of incident cases must be phenotypically tested) (Fig 2A). Given the 555,608 cases of gonorrhea in the US in 2017 [9], the required sampling rate for a 95% probability of detection of a novel variant by the time it occurred in 100 cases would be 16,669 cases per year, or 1,390

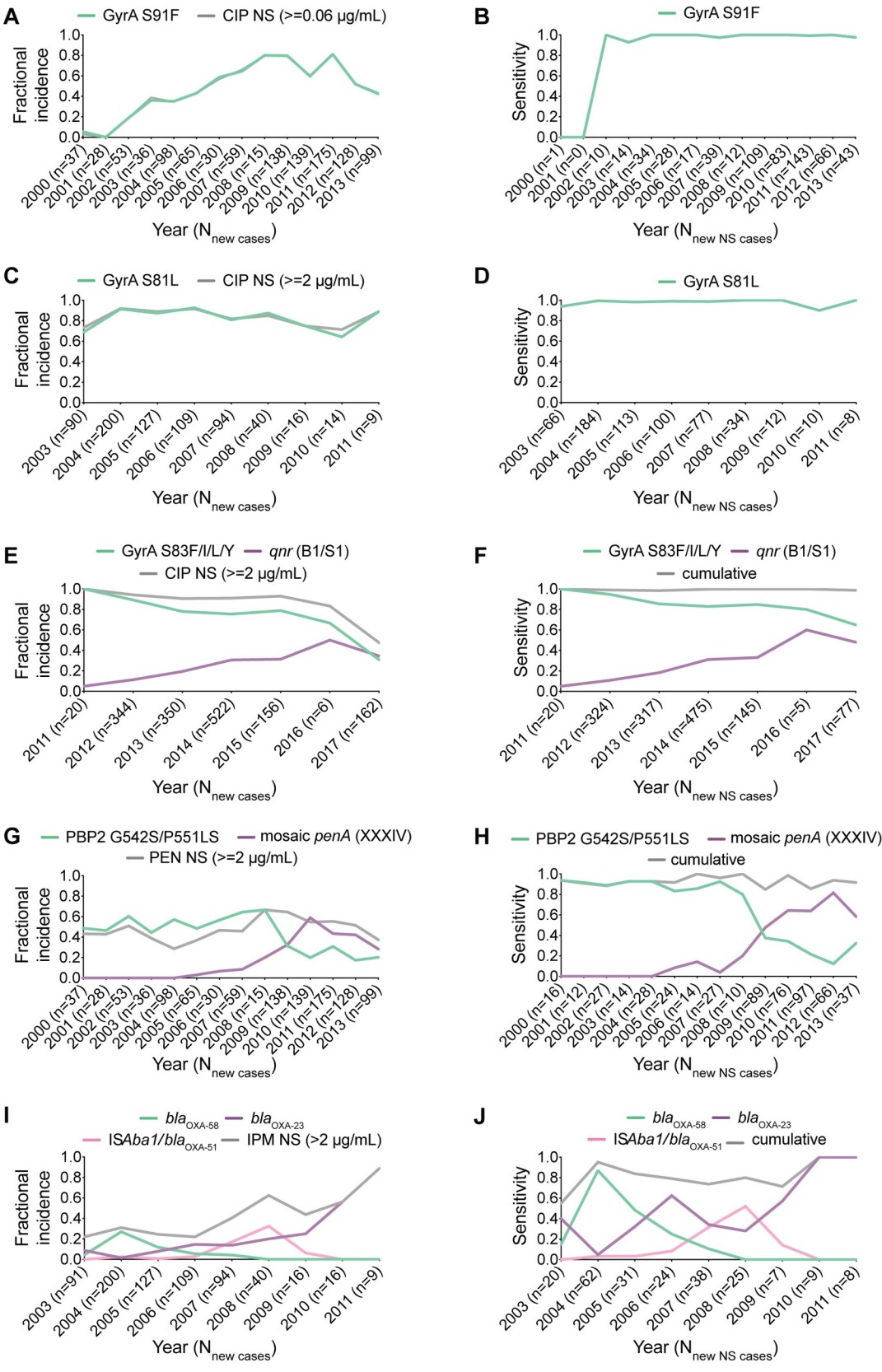

**Fig 1. Emergence of novel resistance variants and their impact on sensitivities of previous variants.** Fractional incidence (A, C, E, G, and I) and sensitivity (B, D, F, H, and J) of genetic variants over time in predicting CIP NS in *N. gonorrhoeae* (A-B), CIP NS in *A. baumannii* (C-D), CIP NS in *K. pneumoniae* (E-F), PEN NS in *N. gonorrhoeae* (G-H), and IPM NS in *A. baumannii* (I-J). Fractional incidence is defined as the proportion of all strains from each year that have the genetic variant or the NS phenotype. Fractional incidence of different markers may not sum to 100% due to uncharacterized resistance markers or strains carrying multiple markers. Sensitivity is defined as the fraction of NS strains from each year that have the genetic variant. Specificity (true negative rate) of variants in predicting NS is not accounted for in these plots. *bla*~OXA~, carbapenem-hydrolyzing class D beta-lactamase gene; CIP, ciprofloxacin; GyrA, DNA gyrase subunit A; IPM, imipenem; IS*Aba*1, *A. baumannii* insertion sequence 1; NS, nonsusceptibility; PBP2, penicillin binding protein 2; PEN, penicillin; *penA*, penicillin binding protein 2 gene; *qnr*, Qnr family pentapeptide repeat protein gene.

cases per month (i.e., *f* = 3% of incident cases). For surveillance programs aimed at detecting novel resistance variants that undermine the sensitivity of a genotype-based diagnostic that has already been implemented in the population, cases with isolates predicted to be resistant by the diagnostic would be excluded from the sampling population, reducing the required sampling rate.

By survival analysis in which the hazard function is defined as the incidence of the novel variant multiplied by the proportion of incident cases that are phenotyped (*f*), if the variant has a growth rate of *r* (i.e., is increasing in fractional incidence [or prevalence, assuming the overall case incidence remains constant] in a population at a rate *r*), then the time (beginning at $t_0$, when the variant first emerged in a single case) at which there is a probability of $1−x$ of having detected the variant (or an *x* probability of having failed to detect the variant) is

$$t = \frac{1}{r} \ln \left( 1 - \frac{r \ln(x)}{f N_0} \right)$$

where $N_0$ is the initial population-wide incidence of the variant in cases.

Based on this model, we can estimate the cost effectiveness of surveillance for genotype–phenotype discordance. We assume surveillance phenotyping is performed on a fraction *f* of all incident cases *I* per unit time such that there is *x* probability of detection of each novel variant by the time *t* that *N* cases of the novel variant have occurred. If the cost of phenotyping an individual isolate is $C_P$, then the total cost from the phenotyping effort required to detect a novel resistance variant is

$$C_{P\ total} = f\ I\ t\ C_P$$

If the cost of a treatment failure is $C_{TF}$, a composite of the costs from the individual clinical failure and secondary cases, the variant occurs in the mean number of expected cases, and assuming that every attempted treatment of infection caused by a pathogen with the variant results in failure, then the expected total cost of treatment failure due to a novel resistance variant is

$$C_{TF\ total} = \frac{C_{TF}}{f}$$

For a pathogen with a given case incidence *I* (e.g., 500,000 cases per year) and phenotyping cost $C_P$ (e.g., US$20 per isolate), the total cost from the phenotyping effort required to detect a novel resistance variant and the total cost from the treatment failures that may be attributed to that novel variant can be determined for a range of assumptions about variant growth rate *r* and cost of treatment failure $C_{TF}$ (**Fig 2B**). Similarly, the cumulative cost associated with phenotyping and treatment failure can be assessed as a function of sampling fraction in order to identify the most cost-effective sampling fraction, defined as that which minimizes the total cost of sampling and treatment failures (**Fig 2C**).

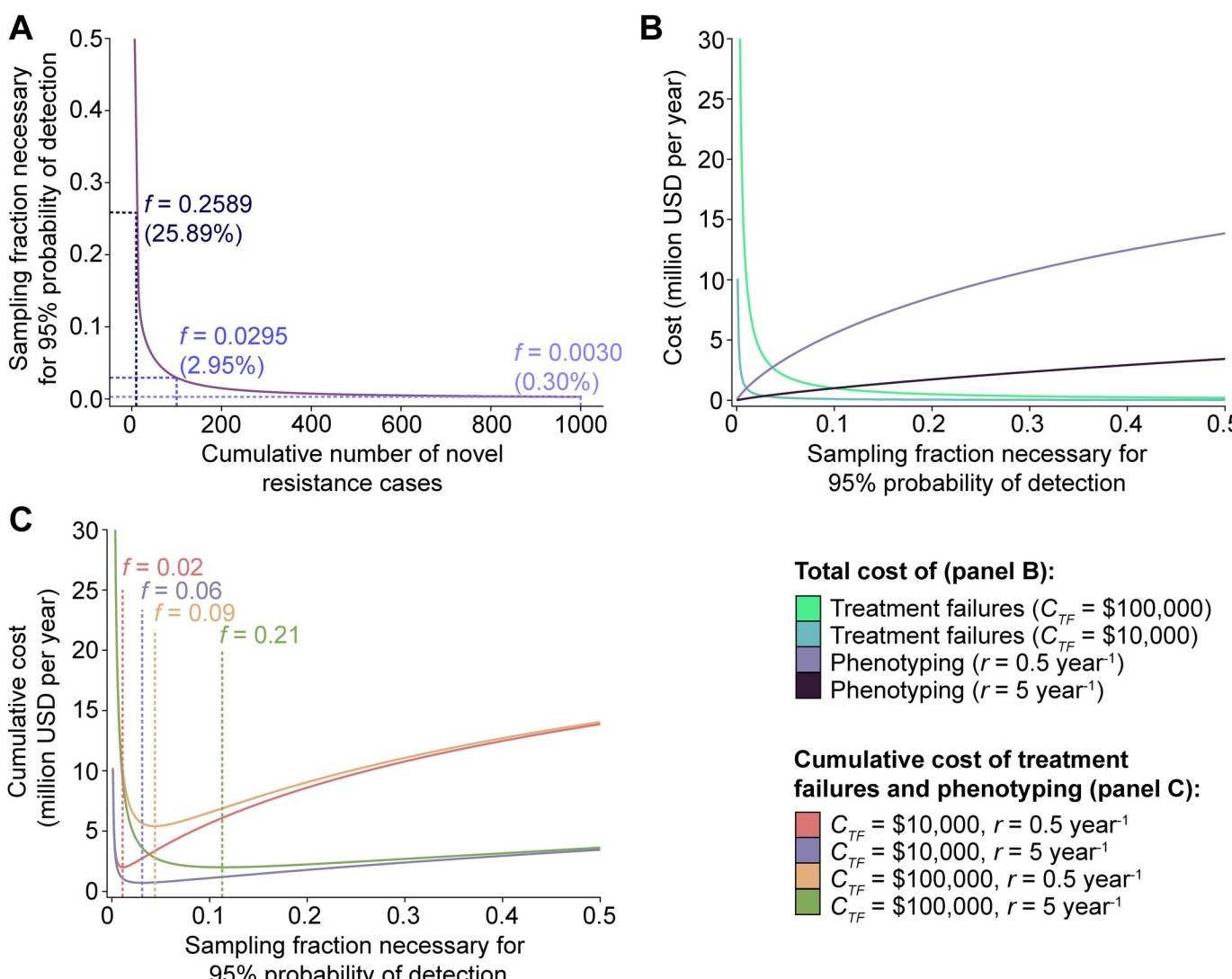

**Fig 2. A framework for the detection of novel resistance variants.** (A) Quantification of sampling fraction (the fraction of incident cases that are phenotyped, $f$) required for 95% probability ($x = 0.95$) of detection of novel variants as a function of the total number of novel resistance cases ($N$) that occur prior to detection. Sampling fractions required for 95% probability of detection of a variant that has occurred in a total of 10, 100, or 1,000 cases are indicated in panel A. (B) Estimation of the total cost associated with the phenotyping required for 95% confidence in detection of a novel resistance variant by the time is has occurred in $N$ cases, assuming an annual case incidence ($I$) of 500,000, a variant growth rate ($r$) of 0.5 or 5 per year, and a phenotyping cost ($C_P$) of US$20 per isolate, and the total cost associated with the mean $N$ expected treatment failures that may be attributed to that novel variant, assuming each individual treatment failure incurs a cost ($C_{TF}$) of US$100,000 or US$10,000. (C) Estimation of the cumulative cost associated with the phenotyping required for 95% confidence in detection of a novel resistance variant by the time it has occurred in $N$ cases and the total cost incurred by the $N$ treatment failures attributed to that novel variant, assuming an annual case incidence ($I$) of 500,000, a variant growth rate ($r$) of 0.5 or 5 per year, a phenotyping cost ($C_P$) of US$20 per isolate, and a cost incurred by each individual treatment failure ($C_{TF}$) of US$100,000 or US$10,000. Sampling fractions ($f$) that minimize the cumulative cost are indicated in C.

## Discussion

Although this sampling model is based on few assumptions and should be generalizable to any resistance variant, the practical implementation of this model requires consideration of multiple additional factors. First, although this model assumes instantaneous testing of isolates, if the phenotyping of the collected clinical isolates is batched, then the intervals between testing could lead to delays in detection. However, the delay is bounded by the selected threshold of allowed failures, the testing interval, and the growth rate of the novel variant in the population.

Second, changes in disease incidence impact the surveillance and sampling strategy. For example, gonorrhea incidence in the US increased 65% between 2008 and 2017 and 18.6% between 2016 and 2017 alone (https://www.cdc.gov/std/stats17). To maintain the same level of confidence that the novel variant will be detected by the desired time or threshold number of cases, disease incidence would need to be closely monitored and surveillance matched accordingly. Given the directly proportional relationship between case incidence and the number of isolates that must be sampled per unit time (sampling rate) to achieve a given objective, an 18.6% increase in incidence of genotypically susceptible strains must correspond to an 18.6% increase in sampling rate in order to maintain the same sampling fraction (*f*) and, thus, the same confidence in detection of a novel variant by the time it has appeared in a given number of cases.

Furthermore, the incidence of clinical isolates predicted to be susceptible (susceptible case incidence), rather than overall case incidence, is of primary relevance for detecting novel resistance determinants. Thus, a third issue to consider in sampling strategy is that the susceptible case incidence may be subject to more rapid changes than overall case incidence, depending on varying selective pressures for or against resistance introduced by a variety of factors, including antibiotic use and the diagnostic itself. Thus, in establishing a plan for a sampling strategy, a conservative approach would be to account for these fluctuations by calculating the necessary sampling rate as a fraction of all cases.

Relatedly, demographic and geographic heterogeneity in selective pressures, and thus in the likelihood of emergence of novel resistance variants, introduces an additional complication in selecting the populations for surveillance and sampling. Behavioral and socioeconomic factors may contribute to differential emergence of antibiotic resistance across subpopulations [10], and certain resistance mechanisms and variants may be more likely to appear in specific subpopulations or transmission networks [6]. For example, for some pathogens, settings such as oncology and critical care units within hospitals, where antibiotic use is highest and where patients who have failed prior antibiotic therapies are likely to concentrate, may provide ideal locations for surveillance. Thus, although a diverse sampling of the population may be optimal in the absence of epidemiological analysis of risk factors for emergence of resistance, the latter may facilitate more targeted sampling strategies that help to reduce delays in detection of novel variants. Similarly, although this model assumes random sampling across a population, it should be noted that this may be difficult to achieve. For example, the Centers for Disease Control and Prevention's Gonococcal Isolate Surveillance Project currently only samples from male patients attending selected sexual health clinics, introducing demographic and geographic bias [11]. Assessment of the impact of demographic and geographic factors on detection efficiency of novel variants may help improve sampling strategies and yield a sampling scheme–tailored model with more estimates.

Delays in updating genotype-based diagnostics may also influence the rates of emergence of new variants because these diagnostics introduce selective pressure against isolates with the diagnostic targets and increased fitness for those lacking the targets [12–14]. Thus, assay adaptability is likely to be an important determinant of diagnostic sustainability. For genotype-based diagnostics that rely on testing for specific alleles, once specimens with unknown pathways to resistance have been identified, it will be important to define the genetic basis of resistance and incorporate it into the diagnostic assay. Thus, long-term support of such diagnostics will require a system for rapidly determining the genetic basis of resistance in novel resistant variants, an activity that is currently challenging for some pathogen species with less tractable genetics and in cases of multifactorial resistance mechanisms. This requirement may create an advantage for diagnostics that rely on phylogenetic similarity [15] and are agnostic to the resistance determinant, for which genetic experiments could be avoided but regularly updating the

reference database will be critical for maintaining sensitivity. However, such approaches are not likely to perform well for drugs for which resistance is frequently gained and lost through de novo mutation and/or horizontal gene transfer and thus are associated with less phylogenetic signal (e.g., as with azithromycin in *N. gonorrhoeae* [8]).

Estimating the costs of expected treatment failures and phenotypic testing as a function of sampling fraction may be useful for identifying the most cost-effective phenotyping rate. However, although published estimates of direct healthcare costs associated with each case of a given infectious disease may serve as a proxy for the cost of treatment failure, such estimates are likely highly variable and will need to be tailored based on factors such as the type of strain (e.g., multidrug-resistant versus extensively drug-resistant *M. tuberculosis*) and the progression of the disease (e.g., uncomplicated gonorrhea versus progression to pelvic inflammatory disease or epididymitis) [16,17]. It will also be important to determine how to incorporate into this estimate indirect costs such as productivity loss, further transmission, or increased antibiotic resistance due to inappropriate use. Furthermore, assessing cost effectiveness requires estimating the rate at which a novel variant can be expected to spread in the population, which may be difficult to reliably predict for all novel variants. However, cost-efficient surveillance may be achieved by tailoring models based on relevant clinical and epidemiological parameters of the pathogen and evaluations of novel variant emergence patterns after implementation of the diagnostic.

This model is based on the assumption that the most efficient and reliable method for detection of novel resistance variants is routine phenotypic testing of strains predicted to be susceptible. However, identification of treatment failures represents an additional and potentially more efficient route to detection [18]. Although the cost-effectiveness framework is based on the assumption that the vast majority of treatment failures will go undetected, depending on factors such as overall case incidence, health system factors, and severity of clinical failure associated with the pathogen, identification of treatment failures may be a more practical alternative to large-scale phenotypic sampling programs. However, identification of treatment failures may be encumbered by a number of factors, including long treatment regimens and/or partial abatement of symptoms and, thus, failure to follow up. Furthermore, infections might be cleared even in the case of undetected resistance, and multidrug therapy may similarly mask novel resistance to individual drugs. For example, one of the first identified cases of infection with the *N. gonorrhoeae* FC428 clone (associated with ceftriaxone resistance and intermediate azithromycin resistance) in the United Kingdom was identified as negative by *N. gonorrhoeae* nucleic acid amplification test (NAAT) 2 weeks after treatment with ceftriaxone and azithromycin, and a second patient in this transmission network showed clinical response to treatment with ceftriaxone and azithromycin before relapse, potentially resulting in transmission to and asymptomatic carriage in her partner [19]. Thus, although continued collection of clinical outcome data is crucial to defining the relationship between phenotypic susceptibility test results and expected treatment outcome, surveillance programs designed to regularly sample a sufficient fraction of isolates in a given population, incorporating relevant epidemiological information, may represent the most reliable strategy for comprehensive detection of novel resistance variants.

## Materials and methods

See **Table 1** for details of the datasets assessed. For all datasets, raw sequence data were downloaded from the NCBI Sequence Read Archive. Genomes were assembled using SPAdes v3.13 [20] with default parameters. Assembly quality was assessed using QUAST v4.3 [21], and contigs <500 bp in length and/or with <10× average coverage were excluded. Antibiotic

**Table 1. Summary of datasets.**

| Species | Dataset description | NS phenotype(s) (associated figure and source) | NCBI SRA Study ID(s) |
|---|---|---|---|
| *N. gonorrhoeae* | Survey from nationwide (US) clinics from 2000 to 2013; male patients only; enriched for ESC and AZM resistance | CIP (**Fig 1A and 1B**), PEN (**Fig 1G and 1H**) [8] | ERP008891, ERP001405, ERP000144 |
| *A. baumannii* | Survey from clinics and hospitals within the US military healthcare system from 2000 to 2012 | CIP (**Fig 1C and 1D**), IPM (**Fig 1I and 1J**) (NCBI BioSample database, BioProject PRJNA300270) | SRP065910 |
| *K. pneumoniae* | Survey from the Houston Methodist hospital system from 2011 to 2017; enriched for β-lactam resistance | CIP (**Fig 1E and 1F**) [7] | SRP102664, SRP110988, SRP116139 |

Abbreviations: AZM, azithromycin; CIP, ciprofloxacin; ESC, extended spectrum cephalosporin; ID, identifier; IPM, imipenem; NCBI SRA, National Center for Biotechnology Information Sequence Read Archive; NS, nonsusceptible; PEN, penicillin

resistance loci were identified in the assembled contigs using BLAST [22], extracted, and aligned using MUSCLE [23] to assess the fractional incidence (the proportion of all isolates from each year that have the variant) and sensitivity (the proportion of all nonsusceptible isolates from each year that have the variant) of resistance variants. Survival analysis was used to relate sampling fractions (the proportion of incident strains receiving confirmatory phenotyping) to the cumulative number of cases of the novel variant prior to detection, the time to detection of the novel variant after emergence, and the cost of phenotyping and treatment failures. The hazard function, or the rate of identifying a strain with the novel variant given that it has not yet been detected, was defined as

$$\lambda(t) = frN_0 e^{(r-\mu)t}$$

where $f$ is the fraction of incident cases phenotyped, $r$ is the growth rate of the novel variant (the rate at which the variant is increasing in fractional incidence [or prevalence, assuming the overall case incidence remains constant] in a population), $N_0$ is the number of cases with the novel variant at the time of emergence (assumed to be 1), $\mu$ is the rate of recovery from infection with a strain with the novel variant (assumed to be $\ll r$, such that $[r-\mu] \sim r$), and $t$ is the time since emergence of the novel variant.

## Author Contributions

**Conceptualization:** Allison L. Hicks, Stephen M. Kissler, Marc Lipsitch, Yonatan H. Grad.

**Data curation:** Allison L. Hicks.

**Formal analysis:** Allison L. Hicks, Stephen M. Kissler, Marc Lipsitch, Yonatan H. Grad.

**Funding acquisition:** Yonatan H. Grad.

**Supervision:** Marc Lipsitch, Yonatan H. Grad.

**Writing – original draft:** Allison L. Hicks.

**Writing – review & editing:** Allison L. Hicks, Stephen M. Kissler, Marc Lipsitch, Yonatan H. Grad.

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
