## [Editor Report · Decision Letter 0]

2 Aug 2019

Dear Dr Grad, 

Thank you for submitting your manuscript entitled "Surveillance to maintain the sensitivity of genotype-based antibiotic resistance diagnostics" for consideration as a Short Reports by PLOS Biology.

Your manuscript has now been evaluated by the PLOS Biology editorial staff as well as by an academic editor with relevant expertise and I am writing to let you know that we would like to send your submission out for external peer review.

*Please be aware that, due to the voluntary nature of our reviewers and academic editors, manuscripts may be subject to delays during the holiday season. Thank you for your patience.*

Please re-submit your manuscript within two working days, i.e. by Aug 04 2019 11:59PM.

Kind regards,

Lauren A Richardson, Ph.D

Senior Editor

PLOS Biology

---

## [Decision Letter · Decision Letter 1]

3 Sep 2019

Dear Dr Grad,

Thank you very much for submitting your manuscript "Surveillance to maintain the sensitivity of genotype-based antibiotic resistance diagnostics" for consideration as a Short Reports at PLOS Biology. Your manuscript has been evaluated by the PLOS Biology editors, an Academic Editor with relevant expertise, and by several independent reviewers.

In light of the reviews (below), we are pleased to offer you the opportunity to address the comments from the reviewers in a revised version that we anticipate should not take you very long. We will then assess your revised manuscript and your response to the reviewers' comments and we may consult the reviewers again.

Of particular note, we encourage you to present this study in a manner more accessible to medical and public health professionals. Rev #3 requests a discussion of resistance mechanisms that may not be captured by this model and Rev #4 questions how the spatial distribution of mutations may impact detection. 

Your revisions should address the specific points made by each reviewer. Please submit a file detailing your responses to the editorial requests and a point-by-point response to all of the reviewers' comments that indicates the changes you have made to the manuscript. In addition to a clean copy of the manuscript, please upload a 'track-changes' version of your manuscript that specifies the edits made. This should be uploaded as a "Related" file type. You should also cite any additional relevant literature that has been published since the original submission and mention any additional citations in your response. 

Before you revise your manuscript, please review the following PLOS policy and formatting requirements checklist PDF: http://journals.plos.org/plosbiology/s/file?id=9411/plos-biology-formatting-checklist.pdf. It is helpful if you format your revision according to our requirements - should your paper subsequently be accepted, this will save time at the acceptance stage.

Please note that as a condition of publication PLOS' data policy (http://journals.plos.org/plosbiology/s/data-availability) requires that you make available all data used to draw the conclusions arrived at in your manuscript. If you have not already done so, you must include any data used in your manuscript either in appropriate repositories, within the body of the manuscript, or as supporting information (N.B. this includes any numerical values that were used to generate graphs, histograms etc.). For an example see here: http://www.plosbiology.org/article/info%3Adoi%2F10.1371%2Fjournal.pbio.1001908#s5.

For manuscripts submitted on or after 1st July 2019, we require the original, uncropped and minimally adjusted images supporting all blot and gel results reported in an article's figures or Supporting Information files. We will require these files before a manuscript can be accepted so please prepare them now, if you have not already uploaded them. Please carefully read our guidelines for how to prepare and upload this data: https://journals.plos.org/plosbiology/s/figures#loc-blot-and-gel-reporting-requirements.

Upon resubmission, the editors assess your revision and assuming the editors and Academic Editor feel that the revised manuscript remains appropriate for the journal, we may send the manuscript for re-review. We aim to consult the same Academic Editor and reviewers for revised manuscripts but may consult others if needed.

We expect to receive your revised manuscript within one month. Please email us (plosbiology@plos.org) to discuss this if you have any questions or concerns, or would like to request an extension. At this stage, your manuscript remains formally under active consideration at our journal; please notify us by email if you do not wish to submit a revision and instead wish to pursue publication elsewhere, so that we may end consideration of the manuscript at PLOS Biology.

When you are ready to submit a revised version of your manuscript, please go to https://www.editorialmanager.com/pbiology/ and log in as an Author. Click the link labelled 'Submissions Needing Revision' where you will find your submission record. 

Sincerely,

Lauren A Richardson, Ph.D

Senior Editor

PLOS Biology

Reviews

Reviewer #1: 

Antibiotic resistance evolution is a significant problem and one way that is currently pursued to tackle the problem is the development of rapid diagnostics. These allow to not only identify the infectious agent, but more importantly the resistance profile and often mutation. A fundamental problem with this, as basically with almost all current approaches to tackle resistance, is that evolutionary change is an ongoing process and this fact is ignored. The paper presented here provides a tool to address this issue: it shows a way to estimate how much testing is required to detect newly evolved resistant variants, given the limits of current tests. As such, this is an interesting and worthwhile approach.

Overall, the paper is well written. I find though, that it lacks in detail and could also be clearer, given that the message is partly addressed at medical professionals with little time to read and digest. 

Lines 48-49. It is no really clear where these data are coming from. In the section on lines 58-73, two publications and NCBI accessions are cited, but it is mostly not clear to which panel in figure 1 they refer. Please clarify the source of the data for each panel of figure 1. 

Figure 1: I think it would be easier to read if fractional incidence and sensitivity were presented on separate panels. This would, especially in panels D and E make it much easier to see the differences between the dashed lines, which contain the main information. Also, I would suggest to briefly explain fractional incidence and sensitivity in the figure legend, to allow readers to understand the figure without going back to the text. 

Lines 82 – 97. The approach is nice and simple. Yet, to reach a wider readership, it could be better explained. First, I would suggest making figure 1F a separate figure with a more informative caption. Also, I would suggest to explain, given the equation, how you arrived at the figures in lines 88 and following, as some readers will not spend much time on the equations. 

Another aspect worth considering here would be that, as also shown in figure 1, different variants might emerge. A brief discussion whether or not that matters would be useful.

---------------

Reviewer #2: 

In this study, the author has filled an important research gap by presenting a mathematical framework to define the sampling rates for confirmatory phenotypic testing so as to detect novel or previously uncommon resistance genotypes. By updating genotype-based diagnostics, the sensitivity of genotype-based antibiotic resistance should therefore maintained high. In addition, the authors also discussed multiple factors that require consideration when using the sampling model. Overall, it is well-structured and written.

Please briefly describe where the datasets were collected from, hospital or other settings, country? 

Line 61-63 As the cumulative sensitivity remains close to 1 in figure 1C and 1D, I am confused about "the genetic markers of resistance show decreased sensitivity over time". I guess the authors meant “For others, the original diagnostic genetic markers of resistance show ….”

Line 100 why the probability of having detected is 1 - X? when x was defined as the probability of new variant will be detected by time N in line 83-84. 

Line 101 Please provide details on how this equation was derived. Is it possible to provide an example like 90-92?

Line 115 How was 3 and 15 additional cases calculated? Was it based on the equation in line 101.

Figure 1A As the fractional incidence in GyrA S91F is a little bit higher than CIP NS in 2000, wondering why the sensitivity for GyrAS91F is 0.

Figure 1F Is X = 0.95?

---------------

Reviewer #3: 

This is a very concise and thoughtful communication on some fundamental aspects of a new (and emerging) method for susceptibility testing. Although the theory is clear, I wonder whether real-life practice might not be (much) more complicated. Resistance can be based on a single genetic event (mutation or allele or gene) as in most of the examples put forward, but also on the combination of multiple events, such as for beta-lactam resistance in Enterobacteriales. For instance the single presence of OXA48 in Klebsiella may still render a susceptible phenotype for imipenem, but addition of any other beta-lactamase may render a non-susceptible phenotype (see Dautzenberg et al, Euro Surveillance 2014 Mar 6;19(9)). Not sure how this would influence the surveillance scheme (and if the authors could elaborate on this).

---------------

Reviewer #4: 

Nicholas G. Davies, signed review

Review of: Surveillance to maintain the sensitivity of genotype-based antibiotic resistance diagnostics

In this manuscript, the authors address an important question for managing antibiotic resistance: how much phenotypic testing for antibiotic resistance is needed to maintain the sensitivity of genotype-based assays for antibiotic resistance?

This is an interesting question with direct implications for policy. The manuscript is accompanied with well-chosen examples illustrating the problem of declining sensitivity of genotype-based diagnostics. There is also a good discussion of the context for the research and of considerations for putting suggestions into practice, as well as of alternative ways to maintain the sensitivity of diagnostics besides surveillance.

At the same time, the main result (line 87) is relatively straightforward to derive, which I think justifies a request that the authors go into a little more detail. For me, there is a slight disconnect here between the practical nature of the problem that is being addressed and the way in which the results are presented.

Specifically, I think the results could be rephrased (or elaborated) to be more relevant to policymakers. While it is interesting to know the required rate of testing, f, such that a new variant is detected with 100x per cent confidence by the time N variant cases have occurred (line 87), policymakers might be more interested in knowing the rate of testing f that maximizes the cost-effectiveness of surveillance, given the cost of testing, the cost of diagnostic failure, the sensitivity of the phenotypic assay for resistance (which may not be 100%, for example if there is a mixed infection), and so on.

Similarly, I’m not sure the time before detection of a novel variant (line 101) is as interesting to policymakers as the expected number of diagnostic failures before detection. Also, as presented, this result depends upon a growth rate r which is probably quite difficult to predict from first principles—after all, we are talking about the relative fitness of novel mutations—and which would no longer be needed by the time it could be measured. Conversely, the number of failures before detection would not depend on r (assuming instantaneous testing).

I have made specific suggestions here but am open to alternatives—just suggesting more generally that the paper would be improved if the results were more directly translatable to decision-making.

Another potential issue that the manuscript doesn’t seem to address is that the model assumes that isolates subjected to phenotypic testing are selected randomly with respect to the overall population being monitored. But the model risks being overconfident if, for example, the relative rate of phenotypic testing varies spatially, since novel mutations will not in general be spread evenly through a population.

Minor issues:

Lines 90-92: The meaning is clear, but there should be a statement about 95% confidence in this sentence.

Lines 98-102: It’s not quite clear from the way this is phrased whether a variant which is remaining at a stable frequency over time should have r = 0 or r = 1. Also, the probability of having detected the variant, which used to be x, now seems to be 1 – x, which is a little confusing. Finally, slightly more detail on how line 101 was derived would be clarifying.

Line 175: the meaning of “identified as NAAT-negative” is a bit opaque—can this be rephrased?

Sincerely,

Nick Davies

London School of Hygiene and Tropical Medicine

---

## [Decision Letter · Decision Letter 2]

16 Oct 2019

Dear Dr Grad,

Thank you for submitting your revised Short Reports entitled "Surveillance to maintain the sensitivity of genotype-based antibiotic resistance diagnostics" for publication in PLOS Biology. I have now obtained advice from three of the original reviewers and have discussed their comments with the Academic Editor. 

As you will read, the reviewers all found your work very well revised. Based on the reviews, we will probably accept this manuscript for publication, assuming that you will modify the manuscript to meet our remaining production requirements. Of note, the manuscript needs and Methods and Materials section.

We expect to receive your revised manuscript within two weeks. Before we will be able to formally accept your manuscript and consider it "in press", we also need to ensure that your article conforms to our guidelines. A member of our team will be in touch shortly with a set of requests. As we can't proceed until these requirements are met, your swift response will help prevent delays to publication.

Please note that you may have the opportunity to make the peer review history publicly available. The record will include editor decision letters (with reviews) and your responses to reviewer comments. If eligible, we will contact you to opt in or out.

Sincerely,

Lauren A Richardson, Ph.D

Senior Editor

PLOS Biology

DATA POLICY:

**Please ensure that figure legends in your manuscript and the Data Statement in the submission system include information on where the underlying data can be found.

For manuscripts submitted on or after 1st July 2019, we require the original, uncropped and minimally adjusted images supporting all blot and gel results reported in an article's figures or Supporting Information files. We will require these files before a manuscript can be accepted so please prepare them now, if you have not already uploaded them. Please carefully read our guidelines for how to prepare and upload this data: https://journals.plos.org/plosbiology/s/figures#loc-blot-and-gel-reporting-requirements.

Reviews

Reviewer #2: 

The manuscript has greatly improved and the authors have addressed the comments/questions that I raised. Thank you.

--------------

Reviewer #3: Yes: Marc Bonten 

No further comments

--------------

Reviewer #4: Yes: Nicholas G. Davies

The authors have satisfactorily addressed my concerns. This paper would make a good contribution to PLOS Biology.

---

## [Editor Report · Decision Letter 3]

29 Oct 2019

Dear Dr Grad,

On behalf of my colleagues and the Academic Editor, Kathryn Elizabeth Holt, I am pleased to inform you that we will be delighted to publish your Short Reports in PLOS Biology. 

Early Version

PRESS 

Kind regards,

Hannah Harwood

Publication Assistant, 

PLOS Biology

on behalf of

Lauren Richardson,

Senior Editor

PLOS Biology